# The relationship between sitting balance, trunk control and mobility with predictive for current mobility level in survivors of sub-acute stroke

Kyeongbong Lee[1], DongGeon Lee[2], SoungKyun Hong[3], DooChul Shin[4], SeYeon Jeong[5], HyeonHui Shin[6], Wonjae Choi [7], SeungHeon An[8], GyuChang Lee [4]*

1 Department of Physical Therapy, Kangwon National University, Samcheok, Republic of Korea,
2 Department of Physical Therapy, Sinsegye Nursing Care Facility, Changwon, Republic of Korea,
3 Department of Physical Therapy, Woosuk University, Wanju, Republic of Korea, 4 Department of Physical Therapy, Kyungnam University, Changwon, Republic of Korea, 5 Department of Physical Therapy, Graduate School of Kyungnam University, Changwon, Republic of Korea, 6 Department of Occupational Therapy, Dongseo University, Busan, Republic of Korea, 7 Department of Physical Therapy, Joongbu University, Geumsan, Republic of Korea, 8 Department of Physical Therapy, National Rehabilitation Center, Seoul, Republic of Korea

* leegc76@kyungnam.ac.kr

**Data Availability Statement:** All relevant data are within the paper.

**Funding:** The authors received no specific funding for this work.

## Abstract

### Objective

To investigate the relationship between sitting balance, trunk control, and mobility, as well as whether the sitting balance and trunk control can predict mobility level in sub-acute stroke survivors.

### Methods

This is a observational and cross-sectional study. Fifty-five hemiplegic stroke survivors were participated in this study. The Timed Up and Go Test (TUG) was used to estimate mobility, and the Sitting Balance Scale (SBS) was used to examining sitting balance. The Trunk Impairment Scale (TIS), Trunk Control Test (TCT), and Postural Assessment Scale for Stroke-trunk control (PASS-TC) were used for examining the trunk control. Spearman's correlation was used to analyze the relationship between TUG, SBS, TIS, TCT, and PASS-TC.

### Results

The TUG is significantly correlated with SBS (r = -0.78), TIS (r = -0.76), TCT (r = -0.65), and PASS-TC (r = -0.67). In addition, the receiver operation characteristic (ROC) curve showed as cut-off value of SBS as >28.5, TIS > 16.5, TCT >82, and PASS-TC >10.5. The area under the ROC curve in each of the four tests is moderately accurate for predicting the mobility of sub-acute stroke survivors (0.84 ~0.90) (0.7 < AUC ≤ 9 (moderate informative)).

**Competing interests:** The authors have declared that no competing interests exist.

## Implications

The SBS showed the highest correlation for mobility using TUG in the hemiplegic stroke survivors. Also, SBS was revealed as the most dominant examination tool predicting the mobility by TUG, it can be explained the sitting postural balance is the variable predicting the mobility in survivors of sub-acute stroke.

## Introduction

Hemiparesis of stroke survivors can reduce the function of trunk and extremities, resulting in impaired sitting and standing balance. Decreased balance ability is a common symptom due to stroke, it can affects gait and activities of daily living [1]. Hence, the balance evaluation of stroke patients is one of the essential factors that can assess the functional level of the stroke patients.

Balance can be measured in a sitting and standing position [2]. Most patients with acute and subacute stroke have a poor sitting balance, they cannot maintain the standing posture. Since, prediction of mobility after stroke is possible to only a few of these patients. However, the mobility is needed for independent daily living, goal setting of rehabilitation through mobility prediction is necessary for all stroke patients. One of the physical goal of stroke rehabilitation is to restore the level of mobility, it is imperative to inform patients and their family of possible levels of mobility recovery to lead a normal social life. In addition, it is needed that prediction of the mobility level can induce active participation in rehabilitation through noticeable motivation. Since, previous studies investigated the prediction for mobility by evaluating post-stroke balance dysfunction. [3–5].

The Berg Balance Scale (BBS) is used to investigate the sitting and standing balance [6], however, BBS may show a floor effect when examining stroke survivors who have decreased balance [7]. In particular, it can be difficult to test the balance of stroke survivors using Likert ranking scales that examine sitting balance [8, 9].

Considering this limitation, Medley et al developed a form of Sitting Balance Scale (SBS) that can examine the patients who have decreased balance and gait abilities [8]. The SBS showed meaningful intra- and inter-rater reliability [8], it is valuable for sitting balance examination in older adults who are non-ambulatory or have limited functional mobility [10]. The SBS appears to be a valid, objective and comprehensive measure of a patient's sitting balance ability.

There are other tools for assessing sitting balance, such as the Trunk Impairment Scale (TIS), Trunk Control Test (TCT), Postural Assessment Scale for Stroke-Trunk Control (PASS-TC). The evaluation tools measure bed mobility, trunk control and sitting balance. The TIS and TCT have a relationship with the following tests for mobility: the gait subscale of Tinneti Performance Oriented Mobility Assessment, Functional Ambulation Category, 10-meter Walk Test in stroke survivors [11]. And the PASS-TC is correlated with Barthel Index and balance subscale of Fugl-Meyer Assessment [12].

As the results of these previous studies, sitting balance, trunk control and mobility are correlated with each other [13–17]. However, if SBS is a tool for evaluating the balance of patients who have remarkably limited gait or have difficulty maintaining the standing balance, it is necessary to verify whether it has discrimination as a sitting balance evaluation tool that can predict whether a person can actually walk.

Therefore, the aim of this study is to confirm the sitting balance and trunk control measured using the evaluation tools, such as SBS, TIS, TCT and PASS-TC, and mobility have

meaningful relationship of sub-acute stroke survivors. In addition, this study is to examine whether the sitting balance and trunk control can predict mobility.

## Methods

### Participants

Sixty-five sub-acute stroke survivors were participated who diagnosed with hemiplegia due to stroke (Between 3 months and 6 months after onset). Inpatient stroke patients recruited through advertisements in the M rehabilitation hospital, and screened according to following criteria. Inclusion criteria were as follows: who have Mini Mental Status Examination > 24, and who are able to walk 5 meters with–or without assistive device [18]. The data was collected from 55 patients and analyzed, due to drop–out caused by health deterioration (3 patients) and discharge (7 patients). All of the participants were noticed the purpose and procedures of the study before they voluntarily signed the consent form.

### Sample size calculation

Using the G Power 3.1 statistical tool to achieve a statistical power of 80% with statistical significance at $p < 0.05$ (two-tailed test) and an effect size of r = 0.5, a total sample size of 26 participants was required for present study.

### Ethics statement

This study was approved by the Kyungnam University Institutional Review Board (Approve number: 1040460-A-2017-013) and conformed to the provisions of the Declaration of Helsinki of 1995.

### Research procedure

General characteristics of participants (age, sex, disease etiology, affected sides, disease duration, and Mini-Mental State Examination scores) were collected through medical chart reviews or brief interviews. And then, all examinations including TUG, SBS, TIS, TCT, and PASS-TC were performed with two research assistants who had experience caring the stroke patients. To verify the difference between sitting balance and trunk control based on the outcome of TUG, the participants were divided into mobility available group (< 20 sec on TUG) and mobility impaired group (≥ 20 sec). In this study, the mobility was examined using the TUG because of objective clinical measure for assessing functional mobility and balance [19]. The SBS for sitting balance, TIS, TCT, and PASS-TC for trunk impairment and control. The participants were allowed to rest for 5–10 min after each examination, following a verbal explanation or demonstration of each test. All examinations were conducted in the following sequence: TUG, SBS, TIS, TCT, and PASS-TC.

### Outcome measurements

**Sitting balance.** *Sitting Balance Scale.* SBS was used to examine the postural sitting balance. This tool form has 11 categories and uses a five-point scale to produce a total score of 40. A higher score results in greater balance ability. The SBS has high intra-rater reliability (ICC = 0.96~0.99) and inter-rater reliability (ICC = 0.87) and internal consistency (*a* = 0.76) [10].

**Trunk control.** *Trunk Impairment Scale.* The TIS consists of three items and scores range from a minimum of 0 to a maximum of 23 points. The items for static postural balance include the ability to cross the unaffected lower extremity over the affected side while maintaining a

seated posture, with both feet on the ground (7 points). The items for dynamic balance (14 points) include the separated movement of the upper and lower extremities through the lateral bending of the trunk. The items for coordination (6 points), include the rotational movements of the scapula and pelvis in the horizontal plane. Inter-rater reliability of TIS was 0.96~0.99 [20].

*Trunk Control Test.* Collin and Wade developed the TCT and it has been widely used for examining the sitting balance in older adults. The TCT is composed of the following four items: rolling to the affected and unaffected side in supine position, supine to sit, and maintaining balance with both feet on the ground for 30 seconds in a seated position. Each item allocated 0 to 25 points with a possible total score of 100. A 0-point in any of these items means the subject cannot perform the movement, while a 12-point score means that the movement was performed in an abnormal pattern. Lastly, a 25-point score means normal movement was achieved. Internal validity of TCT was $a$ = 0.86 at hospitalization and $a$ = 0.83 at discharge, while r = 0.76 (r = 0.70 at hospitalization, r = 0.79 at discharge) [21].

*Postural Assessment Scale for Stroke-Trunk Control.* PASS consists of three postures (supine, sitting, standing); a patient performs 5 items of posture maintenance and 7 items of posture change. Scores range from 0 to 3 points, with a maximum score of 36 points [22]. According to previous studies [9, 20], the five items of trunk control are the following: sitting without support, rolling to the affected side and unaffected side in supine position, sitting on the edge of bed in supine position, and lying in sitting on edge of the bed. This test produces a total score of 15. Inter-rater reliability of PASS-TC was 0.97, and Cronbach $\alpha$ for internal validity $\geq$ 0.93.

**Mobility.**   *Timed Up and Go test.* In this study, the mobility of stroke survivors was assessed by using TUG because it is a simple and quick functional mobility. The TUG was used to examine functional mobility and dynamic balance [23]. It measures the time a person takes to rise from a chair, walk three meters, turn around, walk back to the chair, and sit down. During the test, the person is expected to wear regular footwear and use any mobility assistive device. The test-retest reliability of this test was reported as ICC = 0.96 [24].

## Statistical analysis

Chi square test and Man-Whitney U test was used to compare the general characteristics between mobility available group and mobility impaired group. Spearman's correlation was used to analyze the relationship between TUG, SBS, TIS, TCT, and PASS-TC. ROC curve was used to determine the cut-off value for prediction of mobility. The accuracy of the examination is classified by the area under the ROC curve (AUC), where AUC = 0.5 (non-informative), $0.5 < AUC \leq 7$ (less informative), $0.7 < AUC \leq 9$ (moderate informative), $0.9 < AUC = 1$ (very informative) [25]. In addition, stepwise multivariate regression analysis was used to examine the effect of SBS, TIS, TCT, and PASS-TC on TUG, and the cut-off value was calculated by logistic regression analysis to predict the mobility of two groups. And the odds ratio for estimating the mobility. The significant differences were at $p$-value of $< 0.05$.

## Results

There were no significant differences in age, sex, disease duration, etiology type, affected sides between mobility available group and mobility impaired group, however, in TUG, SBS, TCT, PASS-TC, TIS, there were significant differences between two groups (Table 1).

TUG is significantly correlated with SBS (r = -0.78), TIS (r = -0.76), TCT (r = -0.65), and PASS-TC (r = -0.67) (Table 2).

The ROC curve analysis showed that the SBS cut-off value was > 28.5, TIS > 16.5, TCT > 82, and PASS-TC > 10.5. All four examinations showed that the AUC was a tool for moderate accuracy (0.84 to 0.90, $p < 0.001$) (Table 3).

**Table 1. Participants' characteristics.**

| Variables | Mobility available group (n = 41) | Mobility impaired group (n = 14) | χ2/Z |
|---|---|---|---|
| Age (year) | 54.14 (15.05) | 58.39(12.32) | 0.929 |
| Sex (male/female) | 18 (44%) / 23 (56%) | 9 (64%) / 5 (36%) | 1.735 |
| Disease duration (month) | 4.29 (0.72) | 4.39 (0.62) | 0.429 |
| Etiology type (hemorrhage/ infarction) | 35 (85%) / 6 (15%) | 9 (64%) / 5 (36%) | 2.899 |
| Affected side (Lt./Rt.) | 24 (59%) / 17 (41%) | 7 (50%) / 7 (50%) | 0.309 |
| TUG (score) | 14.64 (2.65) | 33.56 (7.48) | 5.548* |
| SBS (score) | 32.73 (4.47) | 21.36 (8.94) | 3.762* |
| TIS (score) | 17.63 (1.84) | 11.93 (3.71) | 4.527* |
| TCT (score) | 92.76 (10.77) | 74.00 (11.18) | 4.374* |
| PASS-TC (score) | 13.02 (2.23) | 9.50 (2.53) | 3.921* |

NOTE. The characteristics of the participants who met the inclusion criteria for the study is shown.

Values are presented as mean (SD) or n (%).

*p<0.001 as differences between two groups.

TUG < 20 sec as mobility available group, > 20 sec as mobility impaired group.

Abbreviations: TUG, Timed Up & Go test; SBS, Sitting Balance Scale; TIS, Trunk Impairment Scale; TCT, Trunk Control Test; PASS-TC, Postural Assessment Scale for Stroke-Trunk Control.

SBS (β = -0.337) and TIS (β = -0.317) were found to have a 65% effect on the TUG (Table 4). The result from the logistic regression analysis predicted the mobility by using the cut-off value of each examination.

An ROC curve analysis, revealed that SBS (> 28.5 score, odds ratio 0.62) was the most influential variable (Table 5).

## Discussion

The sitting balance and trunk control can be impaired after stroke, it is important to improve it for significant rehabilitation of stroke survivors [11, 17]. In particular, hemiplegic side of trunk can make the impairment of sitting balance and trunk control [26], these factors can limit the mobility of stroke survivors. The results of the present study demonstrated that mobility impaired group had significantly decreased mobility (TUG), sitting balance (SBS), and trunk control (TIS, TCT, PASS-TC) than mobility available group. Trunk stability exercise improve trunk control, dynamic sitting balance, standing balance gait and activities of daily living in subacute stroke survivors [18]. In the results of this study, it was found that the sitting balance and trunk control level were low in patients with subacute stroke with low mobility level by using TUG. If the sitting balance and trunk control level decrease, it is difficult to achieve normal gait because the center of gravity cannot be maintained normally. Thus, the

**Table 2. Correlation between TUG and SBS, TIS, TCT, PASS-TC in participants.**

| | SBS | TIS | TCT | PASS-TC |
|---|---|---|---|---|
| TUG | -0.78* | -0.76* | -0.65* | -0.67* |

*p<0.01 as relationship among variables.

Abbreviations: TUG, Timed Up & Go test; SBS, Sitting Balance Scale; TIS, Trunk Impairment Scale; TCT, Trunk Control Test; PASS-TC, Postural Assessment Scale for Stroke-Trunk Control.

**Table 3. The SBS, TIS, TCT, PASS-TC, cut-off value, sensitivity, specificity, positive/negative predictive values for mobility level in participants.**

| Variables | cut-off value | AUC (95% CI) | p | Sensitivity (%) | Specificity (%) | PPV (%) | NPV (%) |
|---|---|---|---|---|---|---|---|
| SBS | >28.5 (score) | 0.84 (0.68~0.99) | 0.001 | 33/41 (80%) | 11/14 (78%) | 33/36 (91%) | 11/19 (58%) |
| TIS | >16.5 (score) | 0.90 (0.83~0.98) | 0.001 | 32/41 (78%) | 12/14 (85%) | 32/34 (94%) | 12/21 (57%) |
| TCT | >82 (score) | 0.87 (0.78~0.96) | 0.001 | 36/41 (87%) | 10/14 (71%) | 36/40 (90%) | 10/15 (66%) |
| PASS-TC | >10.5 (score) | 0.84 (0.72~0.97) | 0.001 | 36/41 (87%) | 11/14 (78%) | 36/39 (92%) | 11/16 (68%) |

*$p<0.001$ as ROC curve analysis.

TUG < 20 sec as mobility available group, > 20 sec as mobility impaired group.

Abbreviations: AUC, Area Under the Curve; P/NPV, positive/negative predictive values; SBS, Sitting Balance Scale; TIS, Trunk Impairment Scale, TCT, Trunk Control Test; PASS-TC, Postural Assessment Scale for Stroke-Trunk Control.

decreased trunk function makes it difficult to perform normal gait that leads the lowered mobility level.

Through the present study, the SBS demonstrated the highest correlation for mobility using TUG. It is thought the SBS contains the postural balance compared to the other tests, and have items center on the anticipatory demands of upper or lower extremity movement.

The present study, cut-off values of SBS, TCT, PASS-TC and TIS were used as a reference point for estimating the mobility level. In these examinations, it was confirmed that the AUC was moderately accurate (curve area = 0.84 ~ 0.90) [25]. In addition, for the first time, we presented a selection criterion for predicting mobility level. Specificity was acceptable in all four examinations (78% to 87%), specificity (71% to 85%) and positive predictive value (90% to 94%), negative predictive value (57% ~ 68%) was somewhat low.

As a result of this study, prediction of mobility level for stroke survivors who have 4 months of disease duration and minimal to moderate paralysis was found that SBS, TIS, TCT, and PASS-TC are excellent screening methods and show positive predictive value because of sensitivity, specificity and screening test. The negative predictive value of these tests was found to be somewhat low. As in previous studies [10, 23], the study classified that if the time to perform a TUG test is greater than 20 seconds, It's a mobility available group that can walk and if the time to perform a TUG test is equal or less than 20 seconds, it's a mobility impaired group that can't walk. Specifically, the stroke survivors who needed considerably short time for examining TUG, have inappropriate compensatory strategy and use assistive device but independent mobility is possible may not be included. Therefore, more survivors would be required and a more quantitative approach is needed for the classification criteria of mobility level. However, in this study, the selection criteria for discriminating the mobility level in the sitting

**Table 4. Multivariate regression analysis of TUG influence on SBS, TIS, TCT, and PASS-TC results in participants.**

| Variable | Regression coefficient | Standard error | β | t | Revised $r^2$ | F |
|---|---|---|---|---|---|---|
| Constant | 62.729 | 5.061 | | 12.395** | 0.65 | 25.686** |
| SBS | -1.142 | 0.345 | -0.337 | -3.311* | | |
| TIS | -0.387 | 0.132 | -0.317 | -2.940* | | |
| TCT | -0.091 | 0.076 | -0.132 | -1.196 | | |
| PASS-TC | -0.607 | 0.311 | -0.225 | -1.955 | | |

*$p<0.01$

**$p<0.001$ as multiple linear regression analysis.

Response variable: TUG.

Abbreviations: SBS, Sitting Balance Scale; TIS, Trunk Impairment Scale; TCT, Trunk Control Test; PASS-TC, Postural Assessment Scale for Stroke-Trunk Control.

**Table 5. The odds ratio for estimating the mobility level of each variable.**

| Cut-off value (Full scale) | Regression coefficient | Standard error | Wals | Odds ratio | 95% CI | P |
|---|---|---|---|---|---|---|
| SBS >28.5 score (40 score) | -2.789 | 1.119 | 6.209 | 0.62 | 0.007 ∼ 0.551 | 0.013* |
| TIS >16.5 score (23 score) | -2.006 | 1.129 | 3.160 | 0.134 | 0.015 ∼ 1.228 | 0.075 |
| TCT >82 score (100 score) | -1.772 | 1.104 | 2.575 | 0.170 | 0.020 ∼ 1.480 | 0.109 |
| PASS-TC >10.5 score (15 score) | -1.764 | 1.066 | 2.740 | 0.171 | 0.021 ∼ 1.384 | 0.098 |

*$p < 0.05$ as odds ratio.

TUG < 20 sec as mobility available group, > 20 sec as mobility impaired group.

Abbreviations: SBS, Sitting Balance Scale; TIS, Trunk Impairment Scale; TCT, Trunk Control Test; PASS-TC, Postural Assessment Scale for Stroke-Trunk Control.

balance (SBS) and trunk control (TIS, TCT, and PASS-TC) is that sitting balance and trunk control should be maintained before functional movement and restoration of the preceding postural control and mobility [27].

The relationship between SBS and TIS for the elderly was also verified in various conditions (acute phase patient nursing r = 0.92, rehabilitation nursing r = 0.89, intensive nursing r = 0.88, home nursing r = 0.60); both variables are very closely related [10]. In contrast, TCT and PASS-TC consist of only four to five items that examine mobility in bed. Since there is no item that can affect the TUG, it can be seen as a low examination of discrimination tool. In particular, TCT is not capable of qualitative examination of trunk movement [28], and there was a moderate correlation with the trunk muscle strength using the dynamometer [29].

In the present study, it was confirmed that SBS has the highest predictive validity in discriminating mobility level. Unlike TIS, TCT, and PASS-TC, the SBS is consisted of controlling the upper and lower trunk and coordination, as well as a specific task item for examining the comprehensive dynamic balance capability, required for sit to stand task and mobility. This was also confirmed in the factor analysis (except for TCT and PASS-TC) affecting TUG. In the Rasch analysis, static section among TIS items was not suitable because of the ceiling effect in subacute and chronic stroke survivors, this is also seemed to have not affected the power of discrimination of static section [30, 31].

This study have several limitations. First, the sample size in the mobility impaired group was relatively small compared to number of mobility available group. In addition, the age group of the participants was relatively low. And the participants in TUG test were performed based on a specific point in time, thus, it may be limited in its adaptation to all stroke survivors. The second, Medley et al and Thompson reported that SBS is the most objective clinical examination method because it focuses on comprehensive balance examination rather than trunk control examination of TIS [24, 28]. Therefore, in SBS, which is traditionally used, it may be an appropriate examination tool to determine the discrimination of sitting balance and mobility in stroke patients with significantly impaired balance and gait performance. Future studies will need standardization to determine the superiority of sitting balance using convenience sampling of normal individuals and stroke patients. Lastly, the use of new examination tools will require research that reflects the psychological characteristics (sensitivity, specificity, response rate) of patients with various neurological disorders.

## Conclusion

On the present study, the SBS showed the highest correlation for mobility using TUG in sub-acute stroke survivors. In addition, it was found that the SBS is the most appropriate examination tool for predicting the mobility by the TUG in sub-acute stroke survivors. Thus, it may be suggested the sitting balance and trunk control can be important for future mobility. In early

stage of rehabilitation of sub-acute stroke patients, it may be focused on improving the sitting balance and trunk control.

## Author Contributions

**Conceptualization:** DongGeon Lee, SeungHeon An, GyuChang Lee.

**Data curation:** SoungKyun Hong, DooChul Shin, SeungHeon An.

**Formal analysis:** HyeonHui Shin, GyuChang Lee.

**Investigation:** DongGeon Lee, SeungHeon An.

**Methodology:** Kyeongbong Lee, GyuChang Lee.

**Supervision:** GyuChang Lee.

**Writing – original draft:** DongGeon Lee, SeungHeon An, GyuChang Lee.

**Writing – review & editing:** Kyeongbong Lee, SeYeon Jeong, Wonjae Choi, GyuChang Lee.

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
