## [Decision Letter · Decision Letter 0]

23 Oct 2020

PONE-D-20-25714

Correlation among sitting balance, trunk control, mobility, and activities daily living, and predictive validity of mobility level in survivors with sub-acute stroke

PLOS ONE

Dear Dr. Lee,

Thank you for submitting your manuscript to PLOS ONE. After careful consideration, we feel that it has merit but does not fully meet PLOS ONE’s publication criteria as it currently stands. Therefore, we invite you to submit a revised version of the manuscript that addresses the points raised during the review process.

Please address the objections raised by reviewer 2 especially concerning the better explanation of the rationale and the study design. Consider also all the other minor points. 

We look forward to receiving your revised manuscript.

Kind regards,

Andrea Martinuzzi

Academic Editor

PLOS ONE

Journal Requirements:

2. Please address the following:

- Please refer to any sample size calculations performed prior to participant recruitment. If these were not performed please justify the reasons. Please refer to our statistical reporting guidelines for assistance (https://journals.plos.org/plosone/s/submission-guidelines.#loc-statistical-reporting).

- Please include additional information regarding the survey or questionnaire used in the study and ensure that you have provided sufficient details that others could replicate the analyses. For instance, if you developed a questionnaire as part of this study and it is not under a copyright more restrictive than CC-BY, please include a copy, in both the original language and English, as Supporting Information.

3. During your revisions, please note that a simple title correction is required to ensure no errors of grammar: "Correlation between sitting balance, trunk control, mobility and daily living activities with predictive validity of mobility level in survivors of sub-acute stroke". Please ensure this is updated in the manuscript file and the online submission information.

4.Thank you for stating the following financial disclosure:

 [No.].

5.Thank you for stating the following in your Competing Interests section: 

[No.].

6.Please include your tables as part of your main manuscript and remove the individual files. Please note that supplementary tables (should remain/ be uploaded) as separate "supporting information" files

7. Your ethics statement should only appear in the Methods section of your manuscript. If your ethics statement is written in any section besides the Methods, please move it to the Methods section and delete it from any other section. Please ensure that your ethics statement is included in your manuscript, as the ethics statement entered into the online submission form will not be published alongside your manuscript.

Reviewers' comments:

Reviewer's Responses to Questions

**Comments to the Author**

1. Is the manuscript technically sound, and do the data support the conclusions?

Reviewer #1: Partly

Reviewer #2: Yes

2. Has the statistical analysis been performed appropriately and rigorously? 

Reviewer #1: Yes

Reviewer #2: Yes

3. Have the authors made all data underlying the findings in their manuscript fully available?

Reviewer #1: Yes

Reviewer #2: Yes

4. Is the manuscript presented in an intelligible fashion and written in standard English?

Reviewer #1: Yes

Reviewer #2: Yes

5. Review Comments to the Author

Reviewer #1: a. Question 1 → Although the work is very interesting and sound, the introduction is not completely linear and not adequately developed to soundly understand the work.

b. Question 1 → Please, at pag. 9 row 29-30, specify in a more detailed way the aim of the study.

c. Question 4 → Even if the article is totally written in plain and sound English, at pag. 13 rows from 6 to 9, the specific phrase needs to be better explained.

d. Please, at pag. 8 row 10, correct the initial “zero” number as written on all other decimal values in the list.

e. Even if the statistical approach seems sound, at pag. 19, the two groups identified as: “Mobility available group” and “Mobility impaired group” seems to be too much different in population number (41 / 14).

f. Even if it has been declared that age is not a fundamental factor, it seems that all the subject population are young.

Reviewer #2: Review_PONE-D-20-25714

Major comments

#１. What is the significance of using trunk/sitting balance performance to predict mobility?

Please explain in the introduction. While trunk performance and/or sitting balance are important to mobility from previous research, I don’t think that as a reason to exclude other factors (e.g. lower extremity strength). What does reporting moderate predictive validity as a result of limited predictors lead to? Please also highlight the significance of this entire study in the introduction.

#2. Most importantly, I don’t see the clinical significance of predicting current TUG in a patient with measurable TUG. What is important is not predicting TUG, but that the mobility to be assessed by TUG can be explained a lot by sitting balance?

3. The two objectives seem to create confusion. What you want to clarify is the correlation between TUG and other performance test? Or is it trunk/sitting performance as a predictor of TUG? The conclusions do not seem to emphasize the correlation.

4. Although your study is a cross-sectional design, there are several representations that can be used to predict future mobility. Please give appropriate consideration to the information available from the results.

5. What is the significance or hypothesis of the MBI that was evaluated? Do you just want to investigate the correlation to TUG? ADLs were discussed in the previous studies presented in the introduction, but they were not mentioned in the DISCUSSION.

Minor comments

overall

#1. In the title, aim, and abstract, you use “among sitting balance, trunk control, mobility, and activities daily living" and "between sitting balance, trunk control, mobility and ADL”, it's hard to understand what you want to compare them to.

The results only show an correlation between mobility and other factors.

This can lead to a misunderstanding and a misguided purpose, as if it investigated a correlation among all factors.

#2. The key word was “chronic stroke”, is that correct?

abstract

#3. P2, L2. Is it gait ability or mobility that you predict? If you don't distinguish and define it, then please unify.

#4. P2, L10. Is the Finding notation TCT (r = -0.65)? There should be a space before and after “=”.

#5. P2, L11-14. Please indicate what the ROC curve or AUC is intended to do. to predict mobility?

#6. P2, L17. “...predicting the improvement of mobility after stroke.” does not prove improvement from this cross-sectional design.

Introduction

#7. P3, L10-30. The introduction as a whole hypothesizes the usefulness of SBS and these indicators as tools that can predict ADLs. Nevertheless, as a purpose, SBS is not the subject matter and what it predicts is not ADL. So, It’s hard to understand the flow of why you want to predict mobility. The connection from background to purpose seems somewhat forced. Please clarify the hypothesis and revise the structure.

#8. P2, L1-5. The use of the sitting balance tools scarce and BBS is explained at the beginning of the Introduction, but in the context of the “overall, SBS, Trunk Impairment Scale…(P3, L14-16)” there are other assessments that can be measured in the supine and sitting positions besides the SBS, Please explain these discrepancies.

#9. With regard to trunk control and sitting balance used in various contexts, it is often difficult to separate them. For example, the TCT and TIS also include sitting balance as an entry. Are there any definitions used in this study and any differences between there two elements?

Methods

#10. P4, L1-2. There is no mention of walking ability in the subject recruitment, is there a standard? At this time of year, I don't think everyone is capable of TUG. Or should readers understand that only those who are able to walk on advertisement have applied?

#11. P4, L3-4. Related to the above, what are the reasons for drop-outs in cross-sectional studies? Does this mean that there was a request from the patient, but it took a long time for the actual measurement to be taken, resulting in health deterioration and discharge?

#12. P4, L2. What is your rationale for using MMSE > 23 as the inclusion criteria? Does it appear that cognitive ability has a significant impact on the content of this assessment?

#13. P4, L13. Can you indicate the extent of “...experience in using these tools”? It seems ambiguous.

#14. P4, L20-24. Please indicate the original citation for the TUG at the beginning of the description of the TUG (Reference No. 14). In addition, please describe the speed conditions of the TUG.

#15. P4, L28-30. Please reiterate the reference of SBS reliability.

#16. P5, L12-13. Please reiterate the reference of TCT validity.

#17. P5, 34-35. What was the reason for comparing the two groups using only the Chi-square test after confirming the normal distribution in the statistical analysis?

Results

#18. P6, L20. SBS and TIS have similar numbers in the results. What are the correlation coefficients between these two variables?　Discussion also states that it is very closely related [2]. Even though the concepts are different, we get the impression that the content of the test is also partially common. It is recommended to emphasize the concept and relationship between the two variables in the introduction or discussion.

Discussion

#19. P6, L29-30. “The results of the present study...had significantly reduced mobility (TUG),” is natural because it generates two groups according to TUG. It is an unnecessary expression.

#20. P6, L29-34. Several repetitions of the results are described. Please summarize (especially the numbers) and organize them.

#21. P7, L11-13. The sentence of “Specifically, the stroke suvivors ... may not be included.”, what is your rationale for describing the inappropriate compensatory strategy and use assistive device? The authors don't consider the assistive device in recruitment and allow it to be used in the TUG, but what was the device usage rate?

#22. P7, L15-18. What do the authors mean by “However, in this study, ... postural control and mobility." It is somewhat difficult to understand the implications of these sentences.

Is there any evidence to be drawn from the results of this context or from previous studies?”

#23. P7, L19-23. The sentence of “The TCT and ... for TIS, respectively)”, this is the kind of sentence is a repetition of the results, or it should be written in the Results section.

#24. P7, L19. Please state the rationales or add a citation for “the TCT and PASS-TC are suitable for use in acute and subacute patients.”

#25. P7, L23-25. There is a lack of textual basis or citation for “In addition, the balance…required to perform the TUG.”

#26. P8, L4-6. What do these citations explain about “Medley et al and …examination of TIS [15, 18]”? Since SBS and TIS are newer reports than these articles, please clarify what you mean by these citations. Also, what do you mean by “in SBS, which is traditionally used,” in the sentence that follows?

#27. Please make the Limitation clear. As I see it, there are several limitations.

Table 1.

#28. What is the meaning of †?

#29. The percentages of the paralysis location are not shown.

Table 3.

#30. * does not appear to be in the table. Please remove it.

#31. Is this table showing a cut-off score to predict the mobility available group?

I hope that my comment is very useful for the improvement of the article.

6. PLOS authors have the option to publish the peer review history of their article (what does this mean?). If published, this will include your full peer review and any attached files.

Reviewer #1: No

Reviewer #2: No

---

## [Author Response · Author response to Decision Letter 0]

24 Jan 2021

Reply to comments

Reply: Thank you for your kind instruction. As per your comment, we have checked the PLOS ONE style and corrected the manuscript according to the style.

2. Please address the following:

- Please refer to any sample size calculations performed prior to participant recruitment. If these were not performed please justify the reasons. Please refer to our statistical reporting guidelines for assistance (https://journals.plos.org/plosone/s/submission-guidelines.#loc-statistical-reporting).

- Please include additional information regarding the survey or questionnaire used in the study and ensure that you have provided sufficient details that others could replicate the analyses. For instance, if you developed a questionnaire as part of this study and it is not under a copyright more restrictive than CC-BY, please include a copy, in both the original language and English, as Supporting Information.

Reply: Thank you for your valuable comment. As per your comment, we have calculated the sample size and added the description in the Methods section as follows. 

“Sample size calculation

Using the G Power 3.1 statistical tool to achieve a statistical power of 80% with statistical significance at p<0.05 (two-tailed test) and an effect size of r=0.5, a total sample size of 26 participants was required for present study.”

3. During your revisions, please note that a simple title correction is required to ensure no errors of grammar: "Correlation between sitting balance, trunk control, mobility and daily living activities with predictive validity of mobility level in survivors of sub-acute stroke". Please ensure this is updated in the manuscript file and the online submission information.

Reply: Thank you for your valuable comment. As per your comment, we have revised the title as follows.

“Correlation between mobility and sitting postural control, and predictive validity of for mobility level in survivors with sub-acute stroke”

4.Thank you for stating the following financial disclosure:

 [No.].

a. Please clarify the sources of funding (financial or material support) for your study. List the grants or organizations that supported your study, including funding received from your institution.

d. If you did not receive any funding for this study, please state: “The authors received no specific funding for this work.”

Reply: Thank you for your kind instruction. As per your comment, we have added the descriptions as follows.

“Financial disclosure

The authors received no specific funding for this work.

Competing Interests

The authors have declared that no competing interests exist.

Declaration of conflicting interests

The author confirms that there is no conflict of interest.”

[No.].

Reply: Thank you for your kind instruction. As per your comment, we have added the descriptions as follows.

“Financial disclosure

The authors received no specific funding for this work.

Competing Interests

The authors have declared that no competing interests exist.

Declaration of conflicting interests

The author confirms that there is no conflict of interest.”

6. Please include your tables as part of your main manuscript and remove the individual files. Please note that supplementary tables (should remain/ be uploaded) as separate "supporting information" files

Reply: Thank you for your kind instruction. As per your comment, we have included the tables in the manuscript.

7. Your ethics statement should only appear in the Methods section of your manuscript. If your ethics statement is written in any section besides the Methods, please move it to the Methods section and delete it from any other section. Please ensure that your ethics statement is included in your manuscript, as the ethics statement entered into the online submission form will not be published alongside your manuscript.

Reply: Thank you for your kind instruction. As per your comment, we have described the ethic statement in the Methods section only.

Reviewers' comments:

Reviewer's Responses to Questions

Comments to the Author

1. Is the manuscript technically sound, and do the data support the conclusions?

Reviewer #1: Partly

Reviewer #2: Yes

Reply: Thank you for your valuable comment. As per your comment, we have revised the Conclusion section as follows.

“On the present study, the SBS showed the highest correlation for mobility using TUG in sub-acute stroke survivors. Also, SBS was revealed as the most appropriate examination tool for predicting the mobility by TUG in sub-acute stroke survivors. Through additional studies, it should be suggested about standardization to determine the superiority of sitting postural control.”

2. Has the statistical analysis been performed appropriately and rigorously? 

Reviewer #1: Yes

Reviewer #2: Yes

Reply: Thank you for your decision.

3. Have the authors made all data underlying the findings in their manuscript fully available?

Reviewer #1: Yes

Reviewer #2: Yes

Reply: Thank you for your decision.

4. Is the manuscript presented in an intelligible fashion and written in standard English?

Reviewer #1: Yes

Reviewer #2: Yes

Reply: Thank you for your decision.

5. Review Comments to the Author

Reply: Thank you for your kind instruction.

6. PLOS authors have the option to publish the peer review history of their article (what does this mean?). If published, this will include your full peer review and any attached files.

Do you want your identity to be public for this peer review? For information about this choice, including consent withdrawal, please see our Privacy Policy.

Reviewer #1: No

Reviewer #2: No

Reply: Thank you for your kind instruction.

 

Reviewer #1: 

a. Question 1 → Although the work is very interesting and sound, the introduction is not completely linear and not adequately developed to soundly understand the work.

Reply: Thank you for your valuable comment. As per your comment, we have reviewed and revised the Introduction section as follows.

“Introduction

Standardized examination tools that reflect the sitting balance and psychological characteristics are scarce among survivors with significantly lowered balance and gait ability after stroke [1, 2]. In particular, it can be difficult to test the balance of stroke survivors using Likert ranking scales that examine sitting balance [1, 3]. The Berg Balance Scale (BBS) is used to investigate the sitting and standing balance [4], however, BBS may show a floor effect when examining stroke survivors who have decreased balance [5]. Additionally, the Functional Reaching Test, which examines the balance ability with multi-directional movement, is not seen as a meaningful balance examination. This is because scores may be affected by compensatory movement of the unaffected side of trunk [6].

Considering this limitation, Medley et al developed a form of Sitting Balance Scale (SBS) that can examine the patients who have decreased balance and gait abilities [1]. The SBS showed meaningful intra- and inter-rater reliability [1], it is valuable for sitting balance examination in older adults who are non-ambulatory or have limited functional mobility [2]. The SBS appears to be a valid, objective and comprehensive measure of a patient’s sitting balance ability. Also, Trunk Impairment Scale (TIS), Trunk Control Test (TCT), Postural Assessment Scale for Stroke-Trunk Control (PASS-TC) measure bed mobility, trunk control and sitting balance. The TIS and TCT have a relationship with the following tests: the gait subscale of Tinneti Performance Oriented Mobility Assessment (53%∼49%), Functional Ambulation Category (50%∼43%), 10-meter Walk Test (24%∼27%), TUG (36%∼44%) in stroke survivors [7]. The PASS-TC is correlated with BI (r=.89) and balance subscale of Fugl-Meyer Assessment (r=0.73) [8]. 

The normal gait requires a complex interaction of several factors, such as range of motion, muscle strength of lower extremities, nervous system, and trunk function. Timed Up & Go Test (TUG) is one of the most commonly used tools to test balance and mobility in clinical settings. Timed Up & Go test is correlated with gait performance and walking endurance in chronic stroke survivors [9]. 

As the results of these previous studies, sitting balance, trunk control and gait or mobility are correlated with each other, sitting balance [9-14]. However, it is not enough the evidence whether sitting balance and trunk control are related with mobility, and sitting balance and trunk control can predict the mobility level of sub-acute stroke survivors.

Therefore, the aim of this study was to investigate the correlation between sitting balance, trunk control, and mobility, and predictive validity for mobility in sub-acute stroke survivors.”

b. Question 1 → Please, at pag. 9 row 29-30, specify in a more detailed way the aim of the study.

Reply: Thank you for your comment. As per your comment, we have revised the aim of the study as follows.

“Therefore, the aim of this study was to investigate the correlation between sitting balance, trunk control, and mobility, and predictive validity for mobility in sub-acute stroke survivors.”

c. Question 4 → Even if the article is totally written in plain and sound English, at pag. 13 rows from 6 to 9, the specific phrase needs to be better explained.

Reply: Thank you for your detailed comment. The authors reviewed the sentences and made clearer. The changed contents are highlighted in red.

d. Please, at pag. 8 row 10, correct the initial “zero” number as written on all other decimal values in the list.

Reply: Thank you for your detailed comment. The “zero” number was added on all decimal values in the list. 

e. Even if the statistical approach seems sound, at pag. 19, the two groups identified as: “Mobility available group” and “Mobility impaired group” seems to be too much different in population number (41 / 14).

Reply: Thank you for your valuable comment. We totally agree with your opinion. As per your comment, that point was added as one of limitations of this study in Discussion section as follows. 

“This study have several limitations. First, the sample size in mobility impaired group was relatively small compared to number of mobility available group. In addition, the age group of the participants was relatively low. And the participants in TUG test were performed based on a specific point in time, thus, it may be limited in its adaptation to all stroke survivors. The second, Medley et al and Thompson reported that SBS is the most objective clinical examination method because it focuses on comprehensive balance examination rather than trunk control examination of TIS [16, 19]. Therefore, in SBS, which is traditionally used, it may be an appropriate examination tool to determine the discrimination of sitting balance and mobility in stroke patients with significantly impaired balance and gait performance. Future studies will need standardization to determine the superiority of sitting balance using convenience sampling of normal individuals and stroke patients. Lastly, the use of new examination tools will require research that reflects the psychological characteristics (sensitivity, specificity, response rate) of patients with various neurological disorders.”

f. Even if it has been declared that age is not a fundamental factor, it seems that all the subject population are young.

Reply: Thank you for your valuable comment. We totally agree with your opinion. As per your comment, that point was added as one of limitations of this study in Discussion section as follows. 

“This study have several limitations. First, the sample size in mobility impaired group was relatively small compared to number of mobility available group. In addition, the age group of the participants was relatively low. And the participants in TUG test were performed based on a specific point in time, thus, it may be limited in its adaptation to all stroke survivors. The second, Medley et al and Thompson reported that SBS is the most objective clinical examination method because it focuses on comprehensive balance examination rather than trunk control examination of TIS [16, 19]. Therefore, in SBS, which is traditionally used, it may be an appropriate examination tool to determine the discrimination of sitting balance and mobility in stroke patients with significantly impaired balance and gait performance. Future studies will need standardization to determine the superiority of sitting balance using convenience sampling of normal individuals and stroke patients. Lastly, the use of new examination tools will require research that reflects the psychological characteristics (sensitivity, specificity, response rate) of patients with various neurological disorders.”

 

Reviewer #2: Review_PONE-D-20-25714

Major comments

#１. What is the significance of using trunk/sitting balance performance to predict mobility?

Please explain in the introduction. While trunk performance and/or sitting balance are important to mobility from previous research, I don’t think that as a reason to exclude other factors (e.g. lower extremity strength). What does reporting moderate predictive validity as a result of limited predictors lead to? Please also highlight the significance of this entire study in the introduction.

Reply: Thank you for your valuable comment. As per your comment, we have added some description about that and revised the Introduction section as follows.

“Introduction Standardized examination tools that reflect the sitting balance and psychological characteristics are scarce among survivors with significantly lowered balance and gait ability after stroke [1, 2]. In particular, it can be difficult to test the balance of stroke survivors using Likert ranking scales that examine sitting balance [1, 3]. The Berg Balance Scale (BBS) is used to investigate the sitting and standing balance [4], however, BBS may show a floor effect when examining stroke survivors who have decreased balance [5]. Additionally, the Functional Reaching Test, which examines the balance ability with multi-directional movement, is not seen as a meaningful balance examination. This is because scores may be affected by compensatory movement of the unaffected side of trunk [6].

Considering this limitation, Medley et al developed a form of Sitting Balance Scale (SBS) that can examine the patients who have decreased balance and gait abilities [1]. The SBS showed meaningful intra- and inter-rater reliability [1], it is valuable for sitting balance examination in older adults who are non-ambulatory or have limited functional mobility [2]. The SBS appears to be a valid, objective and comprehensive measure of a patient’s sitting balance ability. Also, Trunk Impairment Scale (TIS), Trunk Control Test (TCT), Postural Assessment Scale for Stroke-Trunk Control (PASS-TC) measure bed mobility, trunk control and sitting balance. The TIS and TCT have a relationship with the following tests: the gait subscale of Tinneti Performance Oriented Mobility Assessment (53%∼49%), Functional Ambulation Category (50%∼43%), 10-meter Walk Test (24%∼27%), TUG (36%∼44%) in stroke survivors [7]. The PASS-TC is correlated with BI (r=.89) and balance subscale of Fugl-Meyer Assessment (r=0.73) [8]. 

The normal gait requires a complex interaction of several factors, such as range of motion, muscle strength of lower extremities, nervous system, and trunk function. Timed Up & Go Test (TUG) is one of the most commonly used tools to test balance and mobility in clinical settings. Timed Up & Go test is correlated with gait performance and walking endurance in chronic stroke survivors [9]. 

As the results of these previous studies, sitting balance, trunk control and gait or mobility are correlated with each other, sitting balance [9-14]. However, it is not enough the evidence whether sitting balance and trunk control are related with mobility, and sitting balance and trunk control can predict the mobility level of sub-acute stroke survivors.

Therefore, the aim of this study was to investigate the correlation between sitting balance, trunk control, and mobility, and predictive validity for mobility in sub-acute stroke survivors.”

#2. Most importantly, I don’t see the clinical significance of predicting current TUG in a patient with measurable TUG. What is important is not predicting TUG, but that the mobility to be assessed by TUG can be explained a lot by sitting balance?

Reply: Thank you for your valuable comment. Through this study, we have tried to investigate mainly whether sitting balance and trunk control can have predictive validity for predicting mobility level by TUG. So, we have added description about correlation between TUG and gait performance in chronic stroke patients in Introduction section, as well as another descriptions, as above.

3. The two objectives seem to create confusion. What you want to clarify is the correlation between TUG and other performance test? Or is it trunk/sitting performance as a predictor of TUG? The conclusions do not seem to emphasize the correlation.

Reply: Thank you for your valuable comment. As per your comment, we have revised the aim of the study as follows.

“Therefore, the aim of this study was to investigate the correlation between sitting balance, trunk control, and mobility, and predictive validity for mobility in sub-acute stroke survivors.”

4. Although your study is a cross-sectional design, there are several representations that can be used to predict future mobility. Please give appropriate consideration to the information available from the results.

Reply: Thank you for your valuable comment. As per your comment, for demostrating about that, we have revised the Results section and Discussion section as follows.

“Results

There were no significant differences in age, sex, disease duration, etiology type, affected sides between mobility available group and mobility impaired group, however, in TUG, SBS, TCT, PASS-TC, TIS, there were significant differences between two groups (Table 1). 

TUG is significantly correlated with SBS (r = -0.78), TIS (r = -0.76), TCT (r = -0.65), and PASS-TC (r = -0.67) (Table 2). 

The ROC curve analysis showed that the SBS cut-off value was > 28.5, TIS > 16.5, TCT > 82, and PASS-TC > 10.5. All four examinations showed that the AUC was a tool for moderate accuracy (0.84 to 0.90, p < 0.001) (Table 3). 

SBS (β = -0. 337) and TIS (β = -0. 317) were found to have a 65% effect on the TUG (Table 4). The result from the logistic regression analysis predicted the mobility by using the cut-off value of each examination. 

An ROC curve analysis, revealed that SBS (> 28.5 score, odds ratio 0.62) was the most influential variable (Table 5).

Discussion

The sitting balance and trunk control after stroke can be impaired, and to solve these problems is important for successful rehabilitation of stroke survivors [13, 14]. In particular, impaired sitting balance and trunk control are occurred by hemiplegic side of trunk [23], these factors can limit the functional activities of stroke survivors. The results of the present study demonstrated that mobility impaired group had significantly decreased mobility (TUG), sitting postural balance (SBS), and trunk control (TIS, TCT, PASS-TC) than mobility available group. The mobility by TUG was found to be significantly related to sitting balance (SBS) and trunk control (TIS, TCT, PASS-TC). The present study, cut-off values of SBS, TCT, PASS-TC and TIS were used as a reference point for estimating the mobility level. In these examinations, it was confirmed that the AUC was moderately accurate (curve area = 0.84 ~ 0.90) [22]. In addition, for the first time, we presented a selection criterion for predicting mobility level. Specificity was acceptable in all four examinations (78% to 87%), specificity (71% to 85%) and positive predictive value (90% to 94%), negative predictive value (57% ~ 68%) was somewhat low. 

Through the present study, the SBS demonstrated the highest correlation for mobility using TUG. Also, SBS was revealed as the most dominant examination tool predicting the mobility. It can be explained that the SBS contains the postural balance compared to the other trunk control tests, postural balance can be a meaningful predictor the improvement of mobility in sub-acute stroke survivors. As a result of this study, prediction of mobility level for stroke survivors who have 4 months of disease duration and minimal to moderate paralysis was found that SBS, TIS, TCT, and PASS-TC are excellent screening methods and show positive predictive value because of sensitivity, specificity and screening test. The negative predictive value of these tests was found to be somewhat low. As in previous studies [2, 15], the study classified that if the time to perform a TUG test is greater than 20 seconds, It’s a mobility available group that can walk and if the time to perform a TUG test is equal or less than 20 seconds, it's a mobility impaired group that can’t walk. Specifically, the stroke survivors who needed considerably short time for examining TUG, have inappropriate compensatory strategy and use assistive device but independent mobility is possible may not be included. Therefore, more survivors would be required and a more quantitative approach is needed for the classification criteria of mobility level. However, in this study, the selection criteria for discriminating the mobility level in the sitting balance (SBS) and trunk control (TIS, TCT, and PASS-TC) is that sitting balance and trunk control should be maintained before functional movement and restoration of the preceding postural control and mobility.

TCT and PASS-TC are suitable for use in acute and subacute patients. The TCT and PASS-TC scores of the mobility impaired group in this study were 74 (100 out of 100) and 9.5 (15 out of 12), respectively, but SBS (21.36 points / 40 points) and TIS (11.93 points / 23 points) of mobility impaired group were significantly lower than mobility available group. In addition, the sitting balance and trunk control, capable of moving the predetermined section on the TUG test contents, are necessary components because SBS and TIS examination items are also large part of the tasks required to perform the TUG.

The relationship between SBS and TIS for the elderly was also verified in various conditions (acute phase patient nursing r = 0.92, rehabilitation nursing r = 0.89, intensive nursing r = 0.88, home nursing r = 0.60); both variables are very closely related [2]. In contrast, TCT and PASS-TC consist of only four to five items that examine mobility in bed. Since there is no item that can affect the TUG, it can be seen as a low examination of discrimination tool. In particular, TCT is not capable of qualitative examination of trunk movement [19], and there was a moderate correlation with the trunk muscle strength using the dynamometer [24].

In the present study, it was confirmed that SBS has the highest predictive validity in discriminating mobility level. Unlike TIS, TCT, and PASS-TC, the SBS is consisted of controlling the upper and lower trunk and coordination, as well as a specific task item for examining the comprehensive dynamic balance capability, required for sit to stand task and mobility. This was also confirmed in the factor analysis (except for TCT and PASS-TC) affecting TUG. In the Rasch analysis, static section among TIS items was not suitable because of the ceiling effect in subacute and chronic stroke survivors, this is also seemed to have not affected the power of discrimination of static section [25, 26].

This study have several limitations. First, the sample size in mobility impaired group was relatively small compared to number of mobility available group. In addition, the age group of the participants was relatively low. And the participants in TUG test were performed based on a specific point in time, thus, it may be limited in its adaptation to all stroke survivors. The second, Medley et al and Thompson reported that SBS is the most objective clinical examination method because it focuses on comprehensive balance examination rather than trunk control examination of TIS [16, 19]. Therefore, in SBS, which is traditionally used, it may be an appropriate examination tool to determine the discrimination of sitting balance and mobility in stroke patients with significantly impaired balance and gait performance. Future studies will need standardization to determine the superiority of sitting balance using convenience sampling of normal individuals and stroke patients. Lastly, the use of new examination tools will require research that reflects the psychological characteristics (sensitivity, specificity, response rate) of patients with various neurological disorders.”

5. What is the significance or hypothesis of the MBI that was evaluated? Do you just want to investigate the correlation to TUG? ADLs were discussed in the previous studies presented in the introduction, but they were not mentioned in the DISCUSSION.

Reply: Thank you for your valuable comment. MBI was not main parameter in the study. So, as per your comment, we have deleted related description about MBI.

Minor comments

overall

#1. In the title, aim, and abstract, you use “among sitting balance, trunk control, mobility, and activities daily living" and "between sitting balance, trunk control, mobility and ADL”, it's hard to understand what you want to compare them to.

The results only show an correlation between mobility and other factors.

This can lead to a misunderstanding and a misguided purpose, as if it investigated a correlation among all factors.

Reply: Thank you for your valuable comment. As per your comment, we have revised the title, aim, and abstract of the study as follows.

“Title: Correlation between mobility and sitting postural control, and predictive validity of for mobility level in survivors with sub-acute stroke”

“ABSTRACT

Purpose: To investigate the correlation between sitting balance, trunk control, and mobility, as well as to predict the mobility level in sub-acute stroke survivors.

Design: A observational and cross-sectional study.

Methods: Sixty-five hemiplegic stroke survivors were participated in this study. The Timed Up and Go Test (TUG) was used to estimate mobility, and the Sitting Balance Scale (SBS) was used to examining sitting balance. The Trunk Impairment Scale (TIS), Trunk Control Test (TCT), and Postural Assessment Scale for Stroke-trunk control (PASS-TC) were used for examining the trunk control. 

Findings: The TUG is significantly correlated with SBS (r = -0.78), TIS (r = -0.76), TCT (r = -0.65), and PASS-TC (r = -0.67). In addition, the receiver operation characteristic (ROC) curve showed as cut-off value of SBS as >28.5, TIS > 16.5, TCT >82, and PASS-TC >10.5. The area under the ROC curve in each of the four tests is moderately accurate for predicting the mobility of sub-acute stroke survivors (.84 ~ .90). 

Conclusion: The SBS showed the highest correlation for mobility using TUG in sub-acute stroke survivors. Also, SBS was revealed as the most dominant examination tool predicting the mobility by TUG, it can be explained the sitting postural balance is the variable predicting the mobility in sub-acute stroke survivors.”

“Therefore, the aim of this study was to investigate the correlation between sitting balance, trunk control, and mobility, and predictive validity for mobility in sub-acute stroke survivors.”

#2. The key word was “chronic stroke”, is that correct?

Reply: Thank you for your detailed comment. One of the key word of this study is sub-acute stroke, the “chronic stroke” is incorrect. So, we have revised the expression.

abstract

#3. P2, L2. Is it gait ability or mobility that you predict? If you don't distinguish and define it, then please unify.

Reply: Thank you for your detailed comment. As per your comment, we have deleted the expression ‘gait’ in the Abstract section.

#4. P2, L10. Is the Finding notation TCT (r = -0.65)? There should be a space before and after “=”.

Reply: Thank you for your detailed comment. The authors corrected the notations for insert a space before and after “=”.

#5. P2, L11-14. Please indicate what the ROC curve or AUC is intended to do. to predict mobility?

Reply: Thank you for your detailed comment. As per your comment, we have added the description about that in Abstract as follows.

“In addition, the receiver operation characteristic (ROC) curve showed as cut-off value of SBS as >28.5, TIS > 16.5, TCT >82, and PASS-TC >10.5. The area under the ROC curve in each of the four tests is moderately accurate for predicting the mobility of sub-acute stroke survivors (.84 ~ .90).”

#6. P2, L17. “...predicting the improvement of mobility after stroke.” does not prove improvement from this cross-sectional design.

Reply: Thank you for your detailed comment. The design of the present study is cross-sectional design, so, we deleted the word “improvement” to clarify the conclusion. 

Introduction

#7. P3, L10-30. The introduction as a whole hypothesizes the usefulness of SBS and these indicators as tools that can predict ADLs. Nevertheless, as a purpose, SBS is not the subject matter and what it predicts is not ADL. So, It’s hard to understand the flow of why you want to predict mobility. The connection from background to purpose seems somewhat forced. Please clarify the hypothesis and revise the structure.

Reply: Thank you for your valuable comment. We have revised some description in Introduction section to clarify the aim of this study as follows. 

“Introduction

Standardized examination tools that reflect the sitting balance and psychological characteristics are scarce among survivors with significantly lowered balance and gait ability after stroke [1, 2]. In particular, it can be difficult to test the balance of stroke survivors using Likert ranking scales that examine sitting balance [1, 3]. The Berg Balance Scale (BBS) is used to investigate the sitting and standing balance [4], however, BBS may show a floor effect when examining stroke survivors who have decreased balance [5]. Additionally, the Functional Reaching Test, which examines the balance ability with multi-directional movement, is not seen as a meaningful balance examination. This is because scores may be affected by compensatory movement of the unaffected side of trunk [6].

Considering this limitation, Medley et al developed a form of Sitting Balance Scale (SBS) that can examine the patients who have decreased balance and gait abilities [1]. The SBS showed meaningful intra- and inter-rater reliability [1], it is valuable for sitting balance examination in older adults who are non-ambulatory or have limited functional mobility [2]. The SBS appears to be a valid, objective and comprehensive measure of a patient’s sitting balance ability. Also, Trunk Impairment Scale (TIS), Trunk Control Test (TCT), Postural Assessment Scale for Stroke-Trunk Control (PASS-TC) measure bed mobility, trunk control and sitting balance. The TIS and TCT have a relationship with the following tests: the gait subscale of Tinneti Performance Oriented Mobility Assessment (53%∼49%), Functional Ambulation Category (50%∼43%), 10-meter Walk Test (24%∼27%), TUG (36%∼44%) in stroke survivors [7]. The PASS-TC is correlated with BI (r=.89) and balance subscale of Fugl-Meyer Assessment (r=0.73) [8]. 

The normal gait requires a complex interaction of several factors, such as range of motion, muscle strength of lower extremities, nervous system, and trunk function. Timed Up & Go Test (TUG) is one of the most commonly used tools to test balance and mobility in clinical settings. Timed Up & Go test is correlated with gait performance and walking endurance in chronic stroke survivors [9]. 

As the results of these previous studies, sitting balance, trunk control and gait or mobility are correlated with each other, sitting balance [9-14]. However, it is not enough the evidence whether sitting balance and trunk control are related with mobility, and sitting balance and trunk control can predict the mobility level of sub-acute stroke survivors.

Therefore, the aim of this study was to investigate the correlation between sitting balance, trunk control, and mobility, and predictive validity for mobility in sub-acute stroke survivors.”

#8. P2, L1-5. The use of the sitting balance tools scarce and BBS is explained at the beginning of the Introduction, but in the context of the “overall, SBS, Trunk Impairment Scale…(P3, L14-16)” there are other assessments that can be measured in the supine and sitting positions besides the SBS, Please explain these discrepancies.

Reply: Thank you for your detailed comment. As per your comment, we have revised the Introduction section as above.

#9. With regard to trunk control and sitting balance used in various contexts, it is often difficult to separate them. For example, the TCT and TIS also include sitting balance as an entry. Are there any definitions used in this study and any differences between there two elements?

Reply: Thank you for your detailed comment. We have tried to describe about the two words (sitting balance and trunk control) throughout the text as follows. 

“Title: Correlation between mobility and sitting postural control, and predictive validity of for mobility level in survivors with sub-acute stroke”

“Introduction

Standardized examination tools that reflect the sitting balance and psychological characteristics are scarce among survivors with significantly lowered balance and gait ability after stroke [1, 2]. In particular, it can be difficult to test the balance of stroke survivors using Likert ranking scales that examine sitting balance [1, 3]. The Berg Balance Scale (BBS) is used to investigate the sitting and standing balance [4], however, BBS may show a floor effect when examining stroke survivors who have decreased balance [5]. Additionally, the Functional Reaching Test, which examines the balance ability with multi-directional movement, is not seen as a meaningful balance examination. This is because scores may be affected by compensatory movement of the unaffected side of trunk [6].

Considering this limitation, Medley et al developed a form of Sitting Balance Scale (SBS) that can examine the patients who have decreased balance and gait abilities [1]. The SBS showed meaningful intra- and inter-rater reliability [1], it is valuable for sitting balance examination in older adults who are non-ambulatory or have limited functional mobility [2]. The SBS appears to be a valid, objective and comprehensive measure of a patient’s sitting balance ability. Also, Trunk Impairment Scale (TIS), Trunk Control Test (TCT), Postural Assessment Scale for Stroke-Trunk Control (PASS-TC) measure bed mobility, trunk control and sitting balance. The TIS and TCT have a relationship with the following tests: the gait subscale of Tinneti Performance Oriented Mobility Assessment (53%∼49%), Functional Ambulation Category (50%∼43%), 10-meter Walk Test (24%∼27%), TUG (36%∼44%) in stroke survivors [7]. The PASS-TC is correlated with BI (r=.89) and balance subscale of Fugl-Meyer Assessment (r=0.73) [8]. 

The normal gait requires a complex interaction of several factors, such as range of motion, muscle strength of lower extremities, nervous system, and trunk function. Timed Up & Go Test (TUG) is one of the most commonly used tools to test balance and mobility in clinical settings. Timed Up & Go test is correlated with gait performance and walking endurance in chronic stroke survivors [9]. 

As the results of these previous studies, sitting balance, trunk control and gait or mobility are correlated with each other, sitting balance [9-14]. However, it is not enough the evidence whether sitting balance and trunk control are related with mobility, and sitting balance and trunk control can predict the mobility level of sub-acute stroke survivors.

Therefore, the aim of this study was to investigate the correlation between sitting balance, trunk control, and mobility, and predictive validity for mobility in sub-acute stroke survivors.”

Methods

#10. P4, L1-2. There is no mention of walking ability in the subject recruitment, is there a standard? At this time of year, I don't think everyone is capable of TUG. Or should readers understand that only those who are able to walk on advertisement have applied?

Reply: Thank you for your detailed comment. As per your comment, we have added inclusion criteria as follows.

“Inclusion criteria were as follows: who have Mini Mental Status Examination > 24, and who are able to walk 5meters with – or without assistive device.”

#11. P4, L3-4. Related to the above, what are the reasons for drop-outs in cross-sectional studies? Does this mean that there was a request from the patient, but it took a long time for the actual measurement to be taken, resulting in health deterioration and discharge?

Reply: Thank you for your detailed comment. In fact, the experiment was conducted a few days after the recruitment. In the meantime, applicants who were in poor health or discharged from the hospital were eliminated from the study. That's what it means.

#12. P4, L2. What is your rationale for using MMSE > 23 as the inclusion criteria? Does it appear that cognitive ability has a significant impact on the content of this assessment?

Reply: Thank you for your detailed comment. As per your comment, we have revised the description and added a reference.

#13. P4, L13. Can you indicate the extent of “...experience in using these tools”? It seems ambiguous.

Reply: Thank you for your comment. This sentence seems ambiguous as your advice, we revised the sentence clearer. The changed contents are highlighted in red. 

#14. P4, L20-24. Please indicate the original citation for the TUG at the beginning of the description of the TUG (Reference No. 14). In addition, please describe the speed conditions of the TUG.

Reply: Thank you for your comment. The citation for the TUG corrected in first sentence of the paragraph. The changed contents are highlighted in red.

#15. P4, L28-30. Please reiterate the reference of SBS reliability.

Reply: Thank you for your comment. The reference for the SBS reliability was added in the paragraph. The changed contents are highlighted in red.

#16. P5, L12-13. Please reiterate the reference of TCT validity.

Reply: Thank you for your comment. The reference for the TCT validity was added in the paragraph. The changed contents are highlighted in red.

#17. P5, 34-35. What was the reason for comparing the two groups using only the Chi-square test after confirming the normal distribution in the statistical analysis?

Reply: Thank you for your valuable comment. As per your comment, we have analyzed the nominal data using the Chi-square test, and other data were analyzed by Mann-Whitney U test. We have revised the description as follows. 

“Chi square test and Man-Whitney U test was used to compare the general characteristics between mobility available group and mobility impaired group.”

Results

#18. P6, L20. SBS and TIS have similar numbers in the results. What are the correlation coefficients between these two variables?　Discussion also states that it is very closely related [2]. Even though the concepts are different, we get the impression that the content of the test is also partially common. It is recommended to emphasize the concept and relationship between the two variables in the introduction or discussion.

Reply: Thank you for your valuable comment. As per your comment, we have revised the Introduction section and Discussion section.

Discussion

#19. P6, L29-30. “The results of the present study...had significantly reduced mobility (TUG),” is natural because it generates two groups according to TUG. It is an unnecessary expression.

Reply: Thank you for your comment. As per tour comment, we have reviewed and revised the Discussion section as follows.

“Discussion

The sitting balance and trunk control after stroke can be impaired, and to solve these problems is important for successful rehabilitation of stroke survivors [13, 14]. In particular, impaired sitting balance and trunk control are occurred by hemiplegic side of trunk [23], these factors can limit the functional activities of stroke survivors. The results of the present study demonstrated that mobility impaired group had significantly decreased mobility (TUG), sitting postural balance (SBS), and trunk control (TIS, TCT, PASS-TC) than mobility available group. The mobility by TUG was found to be significantly related to sitting balance (SBS) and trunk control (TIS, TCT, PASS-TC). The present study, cut-off values of SBS, TCT, PASS-TC and TIS were used as a reference point for estimating the mobility level. In these examinations, it was confirmed that the AUC was moderately accurate (curve area = 0.84 ~ 0.90) [22]. In addition, for the first time, we presented a selection criterion for predicting mobility level. Specificity was acceptable in all four examinations (78% to 87%), specificity (71% to 85%) and positive predictive value (90% to 94%), negative predictive value (57% ~ 68%) was somewhat low. 

Through the present study, the SBS demonstrated the highest correlation for mobility using TUG. Also, SBS was revealed as the most dominant examination tool predicting the mobility. It can be explained that the SBS contains the postural balance compared to the other trunk control tests, postural balance can be a meaningful predictor the improvement of mobility in sub-acute stroke survivors. As a result of this study, prediction of mobility level for stroke survivors who have 4 months of disease duration and minimal to moderate paralysis was found that SBS, TIS, TCT, and PASS-TC are excellent screening methods and show positive predictive value because of sensitivity, specificity and screening test. The negative predictive value of these tests was found to be somewhat low. As in previous studies [2, 15], the study classified that if the time to perform a TUG test is greater than 20 seconds, It’s a mobility available group that can walk and if the time to perform a TUG test is equal or less than 20 seconds, it's a mobility impaired group that can’t walk. Specifically, the stroke survivors who needed considerably short time for examining TUG, have inappropriate compensatory strategy and use assistive device but independent mobility is possible may not be included. Therefore, more survivors would be required and a more quantitative approach is needed for the classification criteria of mobility level. However, in this study, the selection criteria for discriminating the mobility level in the sitting balance (SBS) and trunk control (TIS, TCT, and PASS-TC) is that sitting balance and trunk control should be maintained before functional movement and restoration of the preceding postural control and mobility.

TCT and PASS-TC are suitable for use in acute and subacute patients. The TCT and PASS-TC scores of the mobility impaired group in this study were 74 (100 out of 100) and 9.5 (15 out of 12), respectively, but SBS (21.36 points / 40 points) and TIS (11.93 points / 23 points) of mobility impaired group were significantly lower than mobility available group. In addition, the sitting balance and trunk control, capable of moving the predetermined section on the TUG test contents, are necessary components because SBS and TIS examination items are also large part of the tasks required to perform the TUG.

The relationship between SBS and TIS for the elderly was also verified in various conditions (acute phase patient nursing r = 0.92, rehabilitation nursing r = 0.89, intensive nursing r = 0.88, home nursing r = 0.60); both variables are very closely related [2]. In contrast, TCT and PASS-TC consist of only four to five items that examine mobility in bed. Since there is no item that can affect the TUG, it can be seen as a low examination of discrimination tool. In particular, TCT is not capable of qualitative examination of trunk movement [19], and there was a moderate correlation with the trunk muscle strength using the dynamometer [24].

In the present study, it was confirmed that SBS has the highest predictive validity in discriminating mobility level. Unlike TIS, TCT, and PASS-TC, the SBS is consisted of controlling the upper and lower trunk and coordination, as well as a specific task item for examining the comprehensive dynamic balance capability, required for sit to stand task and mobility. This was also confirmed in the factor analysis (except for TCT and PASS-TC) affecting TUG. In the Rasch analysis, static section among TIS items was not suitable because of the ceiling effect in subacute and chronic stroke survivors, this is also seemed to have not affected the power of discrimination of static section [25, 26].

This study have several limitations. First, the sample size in mobility impaired group was relatively small compared to number of mobility available group. In addition, the age group of the participants was relatively low. And the participants in TUG test were performed based on a specific point in time, thus, it may be limited in its adaptation to all stroke survivors. The second, Medley et al and Thompson reported that SBS is the most objective clinical examination method because it focuses on comprehensive balance examination rather than trunk control examination of TIS [16, 19]. Therefore, in SBS, which is traditionally used, it may be an appropriate examination tool to determine the discrimination of sitting balance and mobility in stroke patients with significantly impaired balance and gait performance. Future studies will need standardization to determine the superiority of sitting balance using convenience sampling of normal individuals and stroke patients. Lastly, the use of new examination tools will require research that reflects the psychological characteristics (sensitivity, specificity, response rate) of patients with various neurological disorders.”

#20. P6, L29-34. Several repetitions of the results are described. Please summarize (especially the numbers) and organize them.

Reply: Thank you for your comment. As per your comment, we have reviewed and revised the Discussion section as above.

#21. P7, L11-13. The sentence of “Specifically, the stroke suvivors ... may not be included.”, what is your rationale for describing the inappropriate compensatory strategy and use assistive device? The authors don't consider the assistive device in recruitment and allow it to be used in the TUG, but what was the device usage rate?

Reply: Thank you for your comment. As per your comment, we have reviewed and revised the Discussion section as above.

#22. P7, L15-18. What do the authors mean by “However, in this study,... postural control and mobility." It is somewhat difficult to understand the implications of these sentences.

Is there any evidence to be drawn from the results of this context or from previous studies?”

Reply: Thank you for your comment. As per your comment, we have added the reference for the description.

#23. P7, L19-23. The sentence of “The TCT and ... for TIS, respectively)”, this is the kind of sentence is a repetition of the results, or it should be written in the Results section.

Reply: Thank you for your comment. As per tour comment, we have reviewed and revised the Discussion section as above.

#24. P7, L19. Please state the rationales or add a citation for “the TCT and PASS-TC are suitable for use in acute and subacute patients.”

Reply: Thank you for your comment. As per your comment, we have reviewed and revised the Discussion section as above.

#25. P7, L23-25. There is a lack of textual basis or citation for “In addition, the balance…required to perform the TUG.”

Reply: Thank you for your kind comments. As per your comment, we have reviewed and revised the Discussion section as above.

#26. P8, L4-6. What do these citations explain about “Medley et al and …examination of TIS [15, 18]”? Since SBS and TIS are newer reports than these articles, please clarify what you mean by these citations. Also, what do you mean by “in SBS, which is traditionally used,” in the sentence that follows?

Reply: Thank you for your kind comments. As per your comment, we have reviewed and revised the Discussion section as above.

#27. Please make the Limitation clear. As I see it, there are several limitations.

Reply: Thank you for your valuable comment. As per your comment, at the end of the discussion, the limitations of this study were described in order as follows.

“This study have several limitations. First, the sample size in mobility impaired group was relatively small compared to number of mobility available group. In addition, the age group of the participants was relatively low. And the participants in TUG test were performed based on a specific point in time, thus, it may be limited in its adaptation to all stroke survivors. The second, Medley et al and Thompson reported that SBS is the most objective clinical examination method because it focuses on comprehensive balance examination rather than trunk control examination of TIS [16, 19]. Therefore, in SBS, which is traditionally used, it may be an appropriate examination tool to determine the discrimination of sitting balance and mobility in stroke patients with significantly impaired balance and gait performance. Future studies will need standardization to determine the superiority of sitting balance using convenience sampling of normal individuals and stroke patients. Lastly, the use of new examination tools will require research that reflects the psychological characteristics (sensitivity, specificity, response rate) of patients with various neurological disorders.”

Table 1.

#28. What is the meaning of †?

Reply: Thank you for your detailed comment. As per your comment, we have deleted that.

#29. The percentages of the paralysis location are not shown.

Reply: Thank you for your comment. As per your comment, we have added the percentage s of the paralysis location in the Table 1.

Table 3.

#30. * does not appear to be in the table. Please remove it.

Reply: Thank you for your comment. As per your comment, we have deleted that.

#31. Is this table showing a cut-off score to predict the mobility available group?

Reply: Thank you for your valuable comment. As per your comment, we have deleted that. The receiver operation characteristic (ROC) curve showed as cut-off value of SBS as >28.5, TIS > 16.5, TCT >82, and PASS-TC >10.5. The area under the ROC curve in each of the four tests is moderately accurate for predicting the mobility of sub-acute stroke survivors (.84 ~ .90). It means that if SBS >28.5 points, the probability of predicting or distinguishing the person who can walk is 80% (sensitivity), and if SBS <28.5 points, the possibility of predicting or distinguishing the person who cannot walk is 78%(specificity)

---

## [Decision Letter · Decision Letter 1]

3 Mar 2021

PONE-D-20-25714R1

Correlation between sitting balance, trunk control, mobility and daily living activities with predictive validity of mobility level in survivors of sub-acute stroke

PLOS ONE

Dear Dr. Lee,

Thank you for submitting your manuscript to PLOS ONE. After careful consideration, we feel that it has merit but does not fully meet PLOS ONE’s publication criteria as it currently stands. Therefore, we invite you to submit a revised version of the manuscript that addresses the points raised during the review process.

There are stil comments from the reviewwer 2 that require your attention and need to be addressed properly 

We look forward to receiving your revised manuscript.

Kind regards,

Andrea Martinuzzi

Academic Editor

PLOS ONE

Journal Requirements:

Reviewers' comments:

Reviewer's Responses to Questions

**Comments to the Author**

1. If the authors have adequately addressed your comments raised in a previous round of review and you feel that this manuscript is now acceptable for publication, you may indicate that here to bypass the “Comments to the Author” section, enter your conflict of interest statement in the “Confidential to Editor” section, and submit your "Accept" recommendation.

Reviewer #1: All comments have been addressed

Reviewer #2: (No Response)

2. Is the manuscript technically sound, and do the data support the conclusions?

Reviewer #1: (No Response)

Reviewer #2: Yes

3. Has the statistical analysis been performed appropriately and rigorously? 

Reviewer #1: (No Response)

Reviewer #2: Yes

4. Have the authors made all data underlying the findings in their manuscript fully available?

Reviewer #1: (No Response)

Reviewer #2: Yes

5. Is the manuscript presented in an intelligible fashion and written in standard English?

Reviewer #1: (No Response)

Reviewer #2: Yes

6. Review Comments to the Author

Reviewer #1: (No Response)

Reviewer #2: Review_PONE-D-20-25714

I have checked your revised manuscript, but there are still some things that you cannot responded. Please refer to the following comments to improve your manuscript.

Major comments

#1. What is the significance of using trunk/sitting balance performance to predict mobility? The significance/necessity of this study has not been demonstrated.

The introduction is still not clear and linear. Please clarify the following:

・What is the significance of targeting the sub-acute phase?

・The purpose is not clear. Make it clear whether you want to see relationships or predict mobility. The latter seems to be written like a main message, but if so, it needs to be explicit in the introduction (TUG is the subject in the Findings).

・Please state your research hypothesis. If you want to reveal scale properties in an exploratory manner, please insist on doing so. I don’t understand why mobility is suddenly mentioned at the end of Introduction.

#2. Again, you will also need to answer the following question.

“Most importantly, I don’t see the clinical significance of predicting current TUG in a patient with measurable TUG. What is important is not predicting TUG, but that the mobility to be assessed by TUG can be explained a lot by sitting balance?”

Previous studies have proposed the importance of trunk/sitting balance performance to predict future TUG (not the current TUG).

#3. Related to the above, the overall structure of the introduction section could be further improved. The introduction flow sounds like it emphasizes the advantage of SBS, but is doesn’t seem to be connected to the hypothesis or purpose. Isn’t the flow that SBS is supposed to reflect TUG more than TCT, TIS, or PASS-TC?

If you want to emphasize SBS, the significance of the description of BBS and FRT before it appears to be unimportant. Is SBS an alternative to BBS and FRT? It might be more appropriate to emphasize mobility first instead.

For example.

1. General remarks: stroke and their balance and gait ability

2. Mobility: the significance and necessity of predicting mobility

3. Current status of indices proposed for this purpose

4. Validity of the SBS

5. Hypothesis and purpose

In particular, “2. Mobility“ is important because they determine the significance of this study.

Again, is there a need to bother predicting the current TUG?

If you change the introduction, please reorganize the discussion with the corresponding text.

Minor comments

#4. Title

It may need to be modified to meet the purpose of the text.

Consider whether it is appropriate to use the term "predictive" for current mobility.

For example, it might be as simple as "the relationship between trunk/sitting balance performance and mobility".

#5. Abstract

Your purpose is the same as the title. Isn't the main message to identify what can predict mobility level?　What is the significance of looking at the relationship between clinical indicators?

Is there a specific reason to focus on the sub-acute phase?　If not, this information should be in Methods.

Methods:

The sample size of 65 participants in methods is misleading, and the actual number of participants (55) needs to be indicated. There is no information at all to process the predictions. It is just a list of scales, but it is not a research method.

Findings:

Isn't ROC to “predict the TUG”? Results about predictions are not describe in the methods.

Please standardize the notation of 0 ( .84 or -0.78).

Conclusion:

“Sub-acute stroke survivors” is repeated in the Conclusion. Other redundant expressions are also noticeable

#6. Introduction

You describe the relationship between each indicator, but I don't understand what you are trying to explain. Is the purpose of your research to verify the validity of the scale? It is not appropriate to say that you examined the relationship because it is unknown. Please discuss what you are trying to say by examining the relationship.

#7. Introduction

What does the percentages in the relationship sentences in TIS and TCT stand for?

#8. Introduction

Please do not abbreviate the BI of the first appearance.

#9. Discussion

The discussion is redundant (e.g., repetition of the same terms, repetition of the same kind of arguments). Please organize your arguments.

#10. Discussion

Regarding the sentence "For the first time, we presented a selection criterion for predicting mobility level", as pointed out in the intro, I did not understand the need to show the predicted TUG for those who are capable of TUG using other indicators. Please include this reason and emphasize it again in your discussion. Otherwise, the significance of the study shown “for the first time” will not be conveyed.

#11. Discussion

“It's a mobility available group that can walk and if the time to perform a TUG test is equal or less than 20 seconds, it's a mobility impaired group that can't walk”.

What is the point you are trying to make with these related sentences? Is it a repetition of the methodology?

#12. Conclusion

The conclusion does not receive your purpose. The objective to lead this conclusion is “does the sitting balance correlate with current mobility?” I'm not sure if “prediction” is the right term, but it is common for linear prediction to hold if there is a correlation.

Isn't the basic argument and conclusion not the validity of the predictions, but how the sitting balance relates to mobility?

If you want to argue for predictive validity, I think the title and purpose should reflect “the validity of the sitting balance that predicts the current TUG”.

If the latter, sitting balance is highly correlated with mobility, so it is important to focus on sitting balance in sub-acute stroke.

Additional comment

There are sentences that need to be consistent with the time period as a whole. In particular, the report of previous studies should be shown in the past tense.

7. PLOS authors have the option to publish the peer review history of their article (what does this mean?). If published, this will include your full peer review and any attached files.

Reviewer #1: No

Reviewer #2: No

---

## [Author Response · Author response to Decision Letter 1]

26 Mar 2021

Response to comments

Major comments

#1. What is the significance of using trunk/sitting balance performance to predict mobility? The significance/necessity of this study has not been demonstrated.

The introduction is still not clear and linear. Please clarify the following:

1) What is the significance of targeting the sub-acute phase?

Answer) Thank you for your question. Generally, most stroke patients in acute stage may be still medically unstable and have not started rehabilitation in earnest. And, in chronic stroke patients, recovery was almost advanced, and the recovery rate was slow or nearly reached a plateau. Thus, we investigated for sub-acute stroke patients. Subsequently we tried to verify the relationship between subject's sitting balance and trunk control and mobility, and predictive validity of the sitting balance and trunk control for predicting mobility level.

2) The purpose is not clear. Make it clear whether you want to see relationships or predict mobility. The latter seems to be written like a main message, but if so, it needs to be explicit in the introduction (TUG is the subject in the Findings).

Answer) Thank you for your kind comments. According to your advice, we have rewritten the Introduction section to make it clearer that the order and content of manuscripts have been changed to further emphasize the meaning of SBS. All of changed contents are highlighted in red.

“Introduction

Hemiparesis of stroke survivors can reduce the function of trunk and extremities, resulting in impaired sitting and standing balance. Decreased balance ability is a common symptom due to stroke, it can affects gait and activities of daily living [1]. Hence, the balance evaluation of stroke patients is one of the essential factors that can assess the functional level of the stroke patients.

Balance can be measured in a sitting and standing position [2]. Most patients with acute and subacute stroke have a poor sitting balance, they cannot maintain the standing posture. Since, prediction of mobility after stroke is possible to only a few of these patients. However, the mobility is needed for independent daily living, goal setting of rehabilitation through mobility prediction is necessary for all stroke patients. One of the physical goal of stroke rehabilitation is to restore the level of mobility, it is imperative to inform patients and their family of possible levels of mobility recovery to lead a normal social life. In addition, it is needed that prediction of the mobility level can induce active participation in rehabilitation through noticeable motivation. Since, previous studies investigated the prediction for mobility by evaluating post-stroke balance dysfunction. [3-5].

The Berg Balance Scale (BBS) is used to investigate the sitting and standing balance [6], however, BBS may show a floor effect when examining stroke survivors who have decreased balance [7]. In particular, it can be difficult to test the balance of stroke survivors using Likert ranking scales that examine sitting balance [8, 9].

Considering this limitation, Medley et al developed a form of Sitting Balance Scale (SBS) that can examine the patients who have decreased balance and gait abilities [8]. The SBS showed meaningful intra- and inter-rater reliability [8], it is valuable for sitting balance examination in older adults who are non-ambulatory or have limited functional mobility [10]. The SBS appears to be a valid, objective and comprehensive measure of a patient’s sitting balance ability.

There are other tools for assessing sitting balance, such as the Trunk Impairment Scale (TIS), Trunk Control Test (TCT), Postural Assessment Scale for Stroke-Trunk Control (PASS-TC). The evaluation tools measure bed mobility, trunk control and sitting balance. The TIS and TCT have a relationship with the following tests for mobility: the gait subscale of Tinneti Performance Oriented Mobility Assessment, Functional Ambulation Category, 10-meter Walk Test in stroke survivors [11]. And the PASS-TC is correlated with Barthel Index and balance subscale of Fugl-Meyer Assessment [12]. 

As the results of these previous studies, sitting balance, trunk control and mobility are correlated with each other [13-17]. However, if SBS is a tool for evaluating the balance of patients who have remarkably limited gait or have difficulty maintaining the standing balance, it is necessary to verify whether it has discrimination as a sitting balance evaluation tool that can predict whether a person can actually walk.

Therefore, the aim of this study is to confirm the sitting balance and trunk control measured using the evaluation tools, such as SBS, TIS, TCT and PASS-TC, and mobility have meaningful relationship of sub-acute stroke survivors. In addition, this study is to examine whether the sitting balance and trunk control can predict mobility.”

3) Please state your research hypothesis. If you want to reveal scale properties in an exploratory manner, please insist on doing so. I don’t understand why mobility is suddenly mentioned at the end of Introduction.

Answer) Thank you for your kind comments. The Introduction section was revised why sitting balance and trunk control are important in stroke rehabilitation. Also, sitting balance and trunk control have to be restored to increase the mobility of stroke survivors, and the variances are important to inform patients and their family of possible levels of mobility recovery. All of changed contents are highlighted in red.

#2. Again, you will also need to answer the following question.

1) “Most importantly, I don’t see the clinical significance of predicting current TUG in a patient with measurable TUG. What is important is not predicting TUG, but that the mobility to be assessed by TUG can be explained a lot by sitting balance?”

Previous studies have proposed the importance of trunk/sitting balance performance to predict future TUG (not the current TUG).

Answer) Thank you for your kind comments. We totally agree with your opinion. When comparing the siting balance and trunk mobility based on the current mobility level in Table 1 of this study, it was confirmed that there was a significant difference. In addition, since TUG was used as a variable representing the mobility level of stroke patients in previous studies, mobility was evaluated through TUG assessment in this study. The Introduction section has been revised for a reasonable explanation of this point. All of changed contents are highlighted in red.

#3. Related to the above, the overall structure of the introduction section could be further improved. The introduction flow sounds like it emphasizes the advantage of SBS, but is doesn’t seem to be connected to the hypothesis or purpose. Isn’t the flow that SBS is supposed to reflect TUG more than TCT, TIS, or PASS-TC? 

Answer) Thank you for your kind comments. This study attempted to find the most appropriate evaluation tool to predict mobility by using various methods to assess the siting balance and trunk control. In the rehabilitation process of a stroke patient, predicting the level of mobility that the patient can recover is important for the patient's motivation to participate in rehabilitation and the mental health of the patient's family. According to the results of this study, it was confirmed that the predictability of TUG (mobility) through SBS was the highest compared to TCT, TIS or PASS-TC. We added this point in Introduction section, all of changed contents are highlighted in red.

If you want to emphasize SBS, the significance of the description of BBS and FRT before it appears to be unimportant. Is SBS an alternative to BBS and FRT? It might be more appropriate to emphasize mobility first instead.

For example.

1. General remarks: stroke and their balance and gait ability

2. Mobility: the significance and necessity of predicting mobility

3. Current status of indices proposed for this purpose

4. Validity of the SBS

5. Hypothesis and purpose

Answer) Thank you for your kind comments. First of all, we rewritten the Introduction section again as you suggested. The Introduction section was corrected as the sequence of the subtitle. 

In particular, “2. Mobility“ is important because they determine the significance of this study.

Again, is there a need to bother predicting the current TUG?

If you change the introduction, please reorganize the discussion with the corresponding text.

Answer) Thank you for your kind comments. It is believed that there are many ways to measure the mobility of stroke patients. In this study, however, we assessed the mobility through the TUG because it shows significant validity and reliability in the rehabilitation process of stroke patients. Also, the TUG is a measurement tool that can be used without much difficulty in clinical practice. 

Minor comments

#4. Title

It may need to be modified to meet the purpose of the text.

Consider whether it is appropriate to use the term "predictive" for current mobility.

For example, it might be as simple as "the relationship between trunk/sitting balance performance and mobility".

Answer) Thank you for your kind comments. We corrected the Title of this study as you suggested. All of changed contents are highlighted in red.

“The relationship between sitting balance, trunk control and mobility with predictive for current mobility level in survivors of sub-acute stroke”

#5. Abstract

1) Your purpose is the same as the title. Isn't the main message to identify what can predict mobility level?　What is the significance of looking at the relationship between clinical indicators?

Answer) Thank you for your kind comments. As per your comment, we have revised the description in abstract section. All of changed contents are highlighted in red.

“ABSTRACT

Purpose: To investigate the relationship between sitting balance, trunk control, and mobility, as well as whether the sitting balance and trunk control can predict mobility level in sub-acute stroke survivors.

Design: A observational and cross-sectional study.

Methods: Fifty-five hemiplegic stroke survivors were participated in this study. The Timed Up and Go Test (TUG) was used to estimate mobility, and the Sitting Balance Scale (SBS) was used to examining sitting balance. The Trunk Impairment Scale (TIS), Trunk Control Test (TCT), and Postural Assessment Scale for Stroke-trunk control (PASS-TC) were used for examining the trunk control. 

Findings: The TUG is significantly correlated with SBS (r = -0.78), TIS (r = -0.76), TCT (r = -0.65), and PASS-TC (r = -0.67). In addition, the receiver operation characteristic (ROC) curve showed as cut-off value of SBS as >28.5, TIS > 16.5, TCT >82, and PASS-TC >10.5. The area under the ROC curve in each of the four tests is moderately accurate for predicting the mobility of sub-acute stroke survivors (.84 ~ .90) (0.7 < AUC ≤ 9 (moderate informative)). 

Conclusion: The SBS showed the highest correlation for mobility using TUG in the hemiplegic stroke survivors. Also, SBS was revealed as the most dominant examination tool predicting the mobility by TUG, it can be explained the sitting postural balance is the variable predicting the mobility in survivors of sub-acute stroke.”

2) Is there a specific reason to focus on the sub-acute phase? If not, this information should be in Methods.

Answer) Thank you for your question. Generally, most stroke patients in acute stage may be still medically unstable and have not started rehabilitation in earnest. And, in chronic stroke patients, recovery was almost advanced, and the recovery rate was slow or nearly reached a plateau. Thus, this study investigated for sub-acute stroke patients. Subsequently we tried to verify the relationship between subject's sitting balance and trunk control and mobility, this study was conducted on sub-acute stroke patients. All of changed contents are highlighted in red.

3) Methods:

The sample size of 65 participants in methods is misleading, and the actual number of participants (55) needs to be indicated. There is no information at all to process the predictions. It is just a list of scales, but it is not a research method.

Answer) Thank you for your kind comments. Sixty-five sub-acute stroke survivors were participated and 10 subjects dropped out, thus the results of 55 stroke survivors were analyzed. In Participants section of Method, the changed contents are highlighted in red.

Findings:

Isn't ROC to “predict the TUG”? Results about predictions are not describe in the methods.

Please standardize the notation of 0 ( .84 or -0.78).

Answer) Thank you for your kind comments. As per your comment, we have added the description in methods section of abstract. All of changed contents are highlighted in red.

Conclusion:

“Sub-acute stroke survivors” is repeated in the Conclusion. Other redundant expressions are also noticeable

Answer) Thank you for your kind comments. As per your comment, we have revised the expression. All of changed contents are highlighted in red.

#6. Introduction

You describe the relationship between each indicator, but I don't understand what you are trying to explain. Is the purpose of your research to verify the validity of the scale? It is not appropriate to say that you examined the relationship because it is unknown. Please discuss what you are trying to say by examining the relationship.

Answer) Thank you for your kind comments. As per your comment, we have rewritten the Introduction section to make it clearer that the order and content of manuscripts have been changed to further emphasize the meaning of SBS. All of changed contents are highlighted in red.

#7. Introduction

What does the percentages in the relationship sentences in TIS and TCT stand for?

Answer) Thank you for your kind comments. The percentage is the relationship with mobility assessment tool, such as gait subscale of Tinneti Performance Oriented Mobility Assessment, Functional Ambulation Category, 10-meter Walk Test. We deleted the percentage of each tools to clarify the meaning of the sentences in Introduction. 

#8. Introduction

Please do not abbreviate the BI of the first appearance.

Answer) Thank you for your kind comments. Activities of daily living were deleted due to lack of meaning for the purpose and results of this study. In addition, the order and content of the paragraphs have been revised to clarify the necessity of this study.

#9. Discussion

The discussion is redundant (e.g., repetition of the same terms, repetition of the same kind of arguments). Please organize your arguments.

Answer) Thank you for your kind comments. As per your comments, we have rewritten the discussion section as follows. All of changed contents are highlighted in red.

“Discussion

The sitting balance and trunk control can be impaired after stroke, it is important to improve it for significant rehabilitation of stroke survivors [11, 17]. In particular, hemiplegic side of trunk can make the impairment of sitting balance and trunk control [26], these factors can limit the mobility of stroke survivors. The results of the present study demonstrated that mobility impaired group had significantly decreased mobility (TUG), sitting balance (SBS), and trunk control (TIS, TCT, PASS-TC) than mobility available group. Trunk stability exercise improve trunk control, dynamic sitting balance, standing balance gait and activities of daily living in subacute stroke survivors [18]. In the results of this study, it was found that the sitting balance and trunk control level were low in patients with subacute stroke with low mobility level by using TUG. If the sitting balance and trunk control level decrease, it is difficult to achieve normal gait because the center of gravity cannot be maintained normally. Thus, the decreased trunk function makes it difficult to perform normal gait that leads the lowered mobility level. 

Through the present study, the SBS demonstrated the highest correlation for mobility using TUG. It is thought the SBS contains the postural balance compared to the other tests, and have items center on the anticipatory demands of upper or lower extremity movement. 

The present study, cut-off values of SBS, TCT, PASS-TC and TIS were used as a reference point for estimating the mobility level. In these examinations, it was confirmed that the AUC was moderately accurate (curve area = 0.84 ~ 0.90) [25]. In addition, for the first time, we presented a selection criterion for predicting mobility level. Specificity was acceptable in all four examinations (78% to 87%), specificity (71% to 85%) and positive predictive value (90% to 94%), negative predictive value (57% ~ 68%) was somewhat low. 

As a result of this study, prediction of mobility level for stroke survivors who have 4 months of disease duration and minimal to moderate paralysis was found that SBS, TIS, TCT, and PASS-TC are excellent screening methods and show positive predictive value because of sensitivity, specificity and screening test. The negative predictive value of these tests was found to be somewhat low. As in previous studies [10, 23], the study classified that if the time to perform a TUG test is greater than 20 seconds, It’s a mobility available group that can walk and if the time to perform a TUG test is equal or less than 20 seconds, it's a mobility impaired group that can’t walk. Specifically, the stroke survivors who needed considerably short time for examining TUG, have inappropriate compensatory strategy and use assistive device but independent mobility is possible may not be included. Therefore, more survivors would be required and a more quantitative approach is needed for the classification criteria of mobility level. However, in this study, the selection criteria for discriminating the mobility level in the sitting balance (SBS) and trunk control (TIS, TCT, and PASS-TC) is that sitting balance and trunk control should be maintained before functional movement and restoration of the preceding postural control and mobility [27].

The relationship between SBS and TIS for the elderly was also verified in various conditions (acute phase patient nursing r = 0.92, rehabilitation nursing r = 0.89, intensive nursing r = 0.88, home nursing r = 0.60); both variables are very closely related [10]. In contrast, TCT and PASS-TC consist of only four to five items that examine mobility in bed. Since there is no item that can affect the TUG, it can be seen as a low examination of discrimination tool. In particular, TCT is not capable of qualitative examination of trunk movement [28], and there was a moderate correlation with the trunk muscle strength using the dynamometer [29].

In the present study, it was confirmed that SBS has the highest predictive validity in discriminating mobility level. Unlike TIS, TCT, and PASS-TC, the SBS is consisted of controlling the upper and lower trunk and coordination, as well as a specific task item for examining the comprehensive dynamic balance capability, required for sit to stand task and mobility. This was also confirmed in the factor analysis (except for TCT and PASS-TC) affecting TUG. In the Rasch analysis, static section among TIS items was not suitable because of the ceiling effect in subacute and chronic stroke survivors, this is also seemed to have not affected the power of discrimination of static section [30, 31].

This study have several limitations. First, the sample size in the mobility impaired group was relatively small compared to number of mobility available group. In addition, the age group of the participants was relatively low. And the participants in TUG test were performed based on a specific point in time, thus, it may be limited in its adaptation to all stroke survivors. The second, Medley et al and Thompson reported that SBS is the most objective clinical examination method because it focuses on comprehensive balance examination rather than trunk control examination of TIS [24, 28]. Therefore, in SBS, which is traditionally used, it may be an appropriate examination tool to determine the discrimination of sitting balance and mobility in stroke patients with significantly impaired balance and gait performance. Future studies will need standardization to determine the superiority of sitting balance using convenience sampling of normal individuals and stroke patients. Lastly, the use of new examination tools will require research that reflects the psychological characteristics (sensitivity, specificity, response rate) of patients with various neurological disorders.”

#10. Discussion

Regarding the sentence "For the first time, we presented a selection criterion for predicting mobility level", as pointed out in the intro, I did not understand the need to show the predicted TUG for those who are capable of TUG using other indicators. Please include this reason and emphasize it again in your discussion. Otherwise, the significance of the study shown “for the first time” will not be conveyed.

Answer) Answer) Thank you for your kind comments. As per your comment, we have revised the discussion section as above.

#11. Discussion

“It's a mobility available group that can walk and if the time to perform a TUG test is equal or less than 20 seconds, it's a mobility impaired group that can't walk”.

What is the point you are trying to make with these related sentences? Is it a repetition of the methodology?

Answer) Thank you for your kind comments. It was to find out whether sitting balance and trunk control are correlated with mobility (TUG) as the result of several previous studies. So, the result whether there are differences between sitting balance and trunk control according to mobility level is written in Table 1, and this is described in the Discussion section. 

#12. Conclusion

The conclusion does not receive your purpose. The objective to lead this conclusion is “does the sitting balance correlate with current mobility?” I'm not sure if “prediction” is the right term, but it is common for linear prediction to hold if there is a correlation.

Isn't the basic argument and conclusion not the validity of the predictions, but how the sitting balance relates to mobility?

If you want to argue for predictive validity, I think the title and purpose should reflect “the validity of the sitting balance that predicts the current TUG”.

If the latter, sitting balance is highly correlated with mobility, so it is important to focus on sitting balance in sub-acute stroke.

Answer) Thank you for your kind comments. As per your comment, we have revised the conclusion section as follows.

“Conclusion

On the present study, the SBS showed the highest correlation for mobility using TUG in sub-acute stroke survivors. In addition, it was found that the SBS is the most appropriate examination tool for predicting the mobility by the TUG in sub-acute stroke survivors. Thus, it may be suggested the sitting balance and trunk control can be important for future mobility. In early stage of rehabilitation of sub-acute stroke patients, it may be focused on improving the sitting balance and trunk control.”

Additional comment

There are sentences that need to be consistent with the time period as a whole. In particular, the report of previous studies should be shown in the past tense.

Answer) Thank you for your kind comments. As per your comment, we have checked and revised the whole manuscript.

---

## [Decision Letter · Decision Letter 2]

28 Apr 2021

PONE-D-20-25714R2

The relationship between sitting balance, trunk control and mobility with predictive for current mobility level in survivors of sub-acute stroke

PLOS ONE

Dear Dr. Lee,

Thank you for submitting your manuscript to PLOS ONE. After careful consideration, we feel that it has merit but does not fully meet PLOS ONE’s publication criteria as it currently stands. Therefore, we invite you to submit a revised version of the manuscript that addresses the points raised during the review process.

Please address the remaining suggestion of reviewer 2

We look forward to receiving your revised manuscript.

Kind regards,

Andrea Martinuzzi

Academic Editor

PLOS ONE

Journal Requirements:

Reviewers' comments:

Reviewer's Responses to Questions

**Comments to the Author**

1. If the authors have adequately addressed your comments raised in a previous round of review and you feel that this manuscript is now acceptable for publication, you may indicate that here to bypass the “Comments to the Author” section, enter your conflict of interest statement in the “Confidential to Editor” section, and submit your "Accept" recommendation.

Reviewer #1: All comments have been addressed

Reviewer #2: All comments have been addressed

2. Is the manuscript technically sound, and do the data support the conclusions?

Reviewer #1: Yes

Reviewer #2: Yes

3. Has the statistical analysis been performed appropriately and rigorously? 

Reviewer #1: Yes

Reviewer #2: Yes

4. Have the authors made all data underlying the findings in their manuscript fully available?

Reviewer #1: Yes

Reviewer #2: No

5. Is the manuscript presented in an intelligible fashion and written in standard English?

Reviewer #1: Yes

Reviewer #2: Yes

6. Review Comments to the Author

Reviewer #1: (No Response)

Reviewer #2: Abstract

Is this sufficient to describe the Methods? It doesn't seem to correspond to Finding section.

The following is my previous comment. I will present this again.

Findings:

Isn't ROC to “predict the TUG”? Results about predictions are not describe in the methods.

Please standardize the notation of 0 ( .84 or -0.78).

7. PLOS authors have the option to publish the peer review history of their article (what does this mean?). If published, this will include your full peer review and any attached files.

Reviewer #1: No

Reviewer #2: No

---

## [Author Response · Author response to Decision Letter 2]

4 May 2021

Response to comments

4. Have the authors made all data underlying the findings in their manuscript fully available?

Reviewer #1: Yes

Reviewer #2: No

6. Review Comments to the Author

Reviewer #1: (No Response)

Reviewer #2: Abstract

Is this sufficient to describe the Methods? It doesn't seem to correspond to Finding section.

The following is my previous comment. I will present this again.

Findings:

Isn't ROC to “predict the TUG”? Results about predictions are not describe in the methods.

Please standardize the notation of 0 ( .84 or -0.78).

Answer) Thank you for your kind comments. A sentence corresponding to the result of this study was added in Methods of Abstracts. In addition, we corrected the notations for consistency. We apologize for repeating the same mistake.

---

## [Editor Report · Decision Letter 3]

7 May 2021

The relationship between sitting balance, trunk control and mobility with predictive for current mobility level in survivors of sub-acute stroke

PONE-D-20-25714R3

Dear Dr. Lee,

We’re pleased to inform you that your manuscript has been judged scientifically suitable for publication and will be formally accepted for publication once it meets all outstanding technical requirements.

Kind regards,

Andrea Martinuzzi

Academic Editor

PLOS ONE
---

## [Editor Report · Acceptance letter]

28 Jul 2021

PONE-D-20-25714R3 

The relationship between sitting balance, trunk control and mobility with predictive for current mobility level in survivors of sub-acute stroke 

Dear Dr. Lee:

I'm pleased to inform you that your manuscript has been deemed suitable for publication in PLOS ONE. Congratulations! Your manuscript is now with our production department. 

Kind regards, 

on behalf of

Dr. Andrea Martinuzzi 

Academic Editor

PLOS ONE